# Guided Direct Time-of-Flight Lidar Using Stereo Cameras for Enhanced Laser Power Efficiency

**DOI:** 10.3390/s23218943

**Published:** 2023-11-03

**Authors:** Filip Taneski, Istvan Gyongy, Tarek Al Abbas, Robert K. Henderson

**Affiliations:** 1Institute for Integrated Micro and Nano Systems, University of Edinburgh, Edinburgh EH9 3FF, UKrobert.henderson@ed.ac.uk (R.K.H.); 2Ouster Automotive, Ouster, Inc., Edinburgh EH2 4AD, UK

**Keywords:** lidar, direct time-of-flight, dToF, flash lidar, SPADs, stereo depth, 3D vision

## Abstract

Self-driving vehicles demand efficient and reliable depth-sensing technologies. Lidar, with its capability for long-distance, high-precision measurement, is a crucial component in this pursuit. However, conventional mechanical scanning implementations suffer from reliability, cost, and frame rate limitations. Solid-state lidar solutions have emerged as a promising alternative, but the vast amount of photon data processed and stored using conventional direct time-of-flight (dToF) prevents long-distance sensing unless power-intensive partial histogram approaches are used. In this paper, we introduce a groundbreaking ‘guided’ dToF approach, harnessing external guidance from other onboard sensors to narrow down the depth search space for a power and data-efficient solution. This approach centers around a dToF sensor in which the exposed time window of independent pixels can be dynamically adjusted. We utilize a 64-by-32 macropixel dToF sensor and a pair of vision cameras to provide the guiding depth estimates. Our demonstrator captures a dynamic outdoor scene at 3 fps with distances up to 75 m. Compared to a conventional full histogram approach, on-chip data is reduced by over twenty times, while the total laser cycles in each frame are reduced by at least six times compared to any partial histogram approach. The capability of guided dToF to mitigate multipath reflections is also demonstrated. For self-driving vehicles where a wealth of sensor data is already available, guided dToF opens new possibilities for efficient solid-state lidar.

## 1. Introduction

Self-driving vehicles require a diverse range of depth sensors to ensure safety and reliability [1,2]. This is the consensus among vehicle manufacturers such as Audi, BMW, Ford, and many more, as outlined in their automated driving safety frameworks [3,4]. Sensor types include ultrasound, radar, vision cameras, and lidar. Out of these, lidar can provide long-distance sensing over hundreds of meters with centimeter precision [5]. Direct time-of-flight (dToF), illustrated in Figure 1a, is performed by measuring the roundtrip time of a short laser pulse and is currently the most suited lidar approach for these distances [6]. However, traditional mechanical scanning implementations introduce reliability issues, frame rate limitations, and high cost [7]. For the widespread adoption of self-driving vehicles, a more practical and cost-effective lidar solution is required.

Contrastingly, new solid-state lidar solutions including flash lidar are made using established and economical semiconductor processes with no moving parts. Many solid-state lidar solutions center around a chip containing a 2D array of dToF sensor pixels to time the returning laser from each point in the scene. However, ambient background photons are also present, so the detected arrival times of all photons must be accumulated over multiple laser cycles to distinguish the laser arrival time, as illustrated in Figure 1b. This presents a significant challenge, as each pixel in the dToF sensor must accommodate enough area to detect, process, and time the arrival of photons, as well as store the resulting data. Histogramming, the process of sorting detected photon arrival times into coarse time bins illustrated in Figure 1c, mitigates the challenge of storing large volumes of photon data [8]. However, the requirement to process and store such large volumes of photon data inevitably limits the achievable maximum distance and/or resolution.

The increased adoption of expensive 3D chip stacking processes (Figure 2) to add more histogram bin storage in-pixel demonstrates the value of overcoming this challenge. This is further highlighted by the increased adoption of novel ‘partial histogram’ dToF sensors which concede a limited bin capacity at the cost of greatly increasing lidar laser power consumption (discussed further in Section 2.3).

To overcome the unmanageable volume of photon data, without using power-hungry partial histogram methods, we propose a new ‘guided’ direct time-of-flight approach. Illustrated in Figure 1d, this approach centers around a dToF sensor where the observed time window of each individual pixel can be externally programmed on the fly. By allowing the diverse range of sensors already on the vehicle to guide the sensor, each pixel can efficiently gather the returning laser photons with a reduced set of histogram bins. 

This paper is organized as follows: in Section 2 we discuss related work to highlight the value of the proposed guided dToF approach; in Section 3 the technical details of the implemented guided ToF system are documented; Section 4 presents the achieved performance of the system; and finally a discussion and conclusion of the work is given in Section 5.

## 2. Related Work

Two techniques have so far played a critical role in enabling 2D arrays of dToF pixels for long-range solid-state lidar: 3D chip stacking and partial histogram techniques. This section explores related work in both of these approaches to demonstrate the value and novelty of the proposed guided dToF approach.

### 2.1. 3D Stacked DToF Sensors

While 3D chip stacking has been long established for image sensors, dToF sensors which rely on the high sensitivity and fine time-resolution of single photon avalanche diodes (SPADs) have only been made possible through more recent advancements. The first 3D stacked dToF sensor chip was developed in 2018 by Ximenes et al. [17]. An infinite impulse response (IIR) filter is used, instead of histogramming, to average successive photon arrival times and narrow in on the laser arrival time. However, this technique suffers under ambient conditions, making it impractical for automotive lidar. In 2019, Henderson et al. [10] showcased a stacked dToF sensor with the capacity for 16 histogram bins. They demonstrated ranging outdoors as far as 50 m within an accuracy of tens of centimeters while running at 30 fps, which is a significant step towards automotive-grade depth-sensing performance. In 2021, Padmanabhan et al. [18] highlighted the value of using programmable time windows to achieve long-distance ranging outdoors. The stacked sensor presented achieved a maximum distance of 100 m with 0.7 m error under low ambient light conditions (10 klux), although this is only given for a single-point measurement and at an undisclosed frame rate.

### 2.2. Partial Histogram DToF Sensors

Partial histogram sensors aim to achieve the same maximum range and precision as a full histogram approach (Figure 1c) with a reduced number of histogram bins. They can be grouped into two categories: ‘zooming’ and ‘sliding’. These are illustrated in Figure 3.

Zooming, as illustrated in Figure 3a, spreads the reduced histogram initially across the full distance range. After multiple laser cycles, the peak (signal) bin is identified, and the histogram is zoomed in to a new, narrower time window. Multiple zoom steps can be performed until the required precision is achieved. Zhang et al. [9] published the first dToF sensor capable of independent per-pixel histogram zooming in 2019. Each pixel contained an 8 × 10-bit histogram which zoomed in three steps to achieve a maximum range of 50 m with 8.8 cm accuracy using a 60% reflective target, albeit for a single-point measurement. An updated iteration [14] built on a stacked process enabled an increased histogram bin capacity of 32 bins, allowing zooming to be reduced to a two-step approach. In 2021, Kim et al. [11] reduced the required histogram capacity even further to only two bins using eight zooming steps. The impact of using many zoom steps on frame rate was acknowledged and prompted a follow-up publication by Park et al. [15]. This reduced the number of zoom steps from eight down to four to range up to 33 m at 1.5 fps.

Sliding, as illustrated in Figure 3b, achieves a partial histogram solution by spreading the reduced histogram across only a subset of the full distance range. After sufficient laser cycles have been accumulated, the time window slides to a new time range and the process repeats until the full distance range has been covered. Stoppa et al. [13] published the first sliding histogram sensor in 2021. Using 3D chip stacking, each pixel has the capacity for 32 histogram bins which slide over 16 windows. A total of six bins of overlap between each slide step are used to cover edge cases. The sensor was upgraded the following year [16] using 22 nm technology on the bottom tier (previously 40 nm) to further increase the histogram bin capacity to 59 bins per pixel.

### 2.3. Summary of DToF Histogram Approaches

The conventional full histogram dToF approach efficiently collects returning laser photons, but limited on-chip area makes this approach impractical for long-range, outdoor performance. Even if a full histogram solution could be implemented, the large amount of data output from potentially millions of pixels would only compound the problem of high data volume that self-driving vehicles already face [20].

On the other hand, partial histogram dToF sensors are more feasible. However, these introduce a severe laser power penalty. This occurs in zooming because each step adds an additional set of laser cycles on top of what is required for a full histogram approach, while in sliding, the penalty is a result of most steps not containing the laser return time. This is particularly problematic for flash lidar architectures where a high peak laser power is typically required. The laser power introduced by partial histogram approaches has been extensively studied in [21], showing a minimum 5× laser power penalty is required to meet a typical automotive lidar specification.

In addition to increased laser power, partial histogram approaches introduce other limitations. Sliding does not solve the issue of high data volume and can introduce motion artefacts if the target moves between slide windows within a frame. Zooming can also introduce image artefacts if multiple peaks occur in the same line of sight, e.g., as a result of transparent objects. 

In contrast, if the complexity of integrating multiple onboard data sources can be overcome, a guided dToF system would be able to achieve long-distance outdoor depth sensing with a reduced set of histogram bins and without a laser power penalty. Exploring the feasibility, implementation, and performance of such a system is therefore of significant value in the context of self-driving vehicles, where an abundance of sensor data is already available. A summary of the merits of different dToF histograms alongside the proposed guided dToF approach is provided in Table 1.

## 3. Materials & Methods

This section describes the key elements of the guided dToF system: (i) the guidable dToF sensor and (ii) the source of depth estimation used to guide the lidar, followed by the final integration of the full guided dToF system.

### 3.1. Guided Lidar Sensor

The sensor used to demonstrate guided dToF (Figure 4) was fabricated in a standard 40 nm CMOS technology and features 32 × 64 dToF pixels. Each pixel contains 4 × 4 SPADs alongside processing and storage of photon events into a histogram of 8 × 12-bit bins. Originally presented in [12], each dToF pixel is able to independently slide its histogram time window and automatically lock on to a peak when detected. To use the chip as a guided dToF sensor for this work, the tracking feature has been disabled and configured such that the time window allocated to each pixel can be dynamically programmed.

### 3.2. Guiding Source: Stereo Camera Vision

The variety of sensors available onboard self-driving vehicles, including ultrasound, radar, vision cameras, and geolocation, provide ample data for a guided lidar system. In this work, we use a pair of vision cameras and perform stereo depth estimation to provide the source of guiding. The foundation of stereo depth estimation is to match each point in the image of one (principal) camera to that in the image of another (secondary) camera. The number of pixels any point has shifted by, termed disparity d, gives the distance z to that point according to Equation (1), assuming both cameras are separated by a baseline distance B and share the same focal length f.
(1)$$z=\frac{f\;B}{d}$$

Quantization as a result of discrete pixel disparity values limits the achievable depth accuracy, although sub-pixel estimation can enable resolving disparity to less than a single pixel value [22]. Stereo depth accuracy ∆z is derived in Equation (2), revealing the squared increase in error with distance characteristic of stereo depth.
(2)$$\frac{∆z}{∆d}=-\frac{f\;B}{d^{2}}\Rightarrow \left|∆z\right|=\frac{z^{2}\;∆d}{f\;B}\;$$

In reality, the achieved accuracy is limited by the point-matching ability of the chosen stereo processing algorithm [23]. Although state-of-the-art machine learning algorithms now outperform traditional computer vision algorithms for stereo depth estimation [24], the aim of this work is to prove the concept of guided lidar. Therefore, we adopt the established semi-global matching (SGM) algorithm for simplicity [25]. Figure 5 shows the process used to acquire stereo depth estimates in our guided dToF system.

Prior to running, the cameras must be carefully calibrated by imaging a checkerboard in various poses [26]. This allows the intrinsic (focal length and optical center) and extrinsic (relative translation and rotation) camera parameters to be extracted. These are used during runtime for both rectification and conversion of disparity to distance. Rectification allows the stereo matching search space to be dramatically reduced by aligning all points in both images along the same horizontal plane.

### 3.3. Pixel Mapping

Once depth estimates of the scene have been acquired, they must be mapped onto each individual pixel of the lidar sensor to guide it to the appropriate depth window. The process of mapping a depth estimate from a pixel in the principal (left) stereo camera to the lidar sensor is illustrated in Figure 6. Camera calibration is once again adopted to determine the translation of the lidar sensor with respect to the principal stereo camera. Capturing checkerboard images using the lidar sensor is achieved by configuring it for photon counting using intensity data. After calibration, the parameters required for pixel mapping are established: the intrinsic matrix of the principal stereo camera $K_{s}$, the intrinsic matrix of the lidar $K_{l}$, and the extrinsic parameters of the lidar with respect to the principal camera position, composed of rotation matrix $R_{l}$ and translation matrix $T_{l}$. 

The pixel mapping process is achieved in two steps: (a) map each pixel coordinate in the stereo depth image ($x_{s},y_{s}$) to its corresponding world coordinate (X,Y,Z), and then (b) map each world coordinate to the corresponding lidar camera pixel coordinate ($x_{l},y_{l}$). The first step is achieved by multiplying the inverse intrinsic matrix of the principal stereo camera $K_{s}$ by the camera coordinate to give a normalized world coordinate.
(3)$$K_{s}^{-1}\times \left(\begin{array}{c} x_{s}\\ y_{s}\\ 1 \end{array}\right)=\left(\begin{array}{c} \tilde{X}\\ \tilde{Y}\\ 1 \end{array}\right)$$

The world coordinate can then be scaled appropriately by multiplying by the distance Z to that point, estimated prior by the stereo depth algorithm.
(4)$$Z∙\left(\begin{array}{c} \tilde{X}\\ \tilde{Y}\\ 1 \end{array}\right)=\left(\begin{array}{c} X\\ Y\\ Z \end{array}\right)$$

The second step is achieved by multiplying the lidar camera’s extrinsic matrix $P_{l}$ by the prior calculated world coordinated, giving the corresponding lidar camera coordinate.
(5)$$P_{l}\times \left(\begin{array}{c} X\\ Y\\ \begin{array}{c} Z\\ 1 \end{array} \end{array}\right)=\left(\begin{array}{c} x_{l}\\ y_{l}\\ 1 \end{array}\right)$$
where $P_{l}=K_{l}\times \left[R_{l}T_{l}\right]$. 

In the case where multiple camera pixels map to one lidar pixel, the modal pixel value can be taken. Alternatively, duplicates may be discarded to save processing time. 

### 3.4. Process Optimization

The stereo depth and pixel matching processes can be greatly optimized by reducing the total data processed in the pipeline. Figure 7a shows a full 1080 × 1440 resolution image produced by one of the stereo vision cameras. Figure 7b shows the total processing time required to rectify and run the SGM algorithm in our setup is approximately 0.4 s, equivalent to a maximum frame rate of around 2 fps. However, the projected lidar field-of-view overlaps only a small portion of the camera image. In addition, multiple pixels of the stereo vision camera occupy a single projected lidar pixel. By acquiring images cropped to half height and enabling pixel binning in 2 × 2, the total amount of data is reduced eight-fold, reducing the stereo depth processing time to 50 ms without degrading the guiding depth estimates. Moreover, this also reduces the number of coordinates that need to be point-matched to the lidar sensor.

### 3.5. Process Flow

An overview of each process step in the implemented guided dToF lidar system is illustrated in Figure 8. After acquiring images from the stereo cameras, a stereo depth estimate image is established from the perspective of the chosen principal (left) camera. Depth estimates are then mapped to each pixel of the lidar sensor as described in Section 3.3. The exposure time window of each lidar pixel is then programmed to the interval corresponding to the provided depth estimate. Finally, the lidar acquisition period begins, with each pixel building a histogram of photon returns within its allocated time window to converge on a precisely measured distance, producing a depth map.

### 3.6. Software

For this implementation, each step of the guided dToF process is programmed in MATLAB [27] (R2023b; MathWorks; Natick, MA, USA.) running on a 1.9 GHz Intel Core i7 8th generation laptop. The image acquisition toolbox is used to acquire images from the stereo cameras. The computer vision toolbox is used to perform the rectification, implement the chosen stereo matching algorithm (SGM), and convert disparity to distance. The Stereo Camera Calibrator app also contained in this toolbox is used to perform checkerboard calibration and extract camera parameters.

### 3.7. Setup

Details of the guided dToF lidar demonstrator are presented in Table 2. The lidar sensor bin widths are configured to 0.39 m (2.6 ns) as an optimum ratio to the laser pulse width as recommended in [28]. While many solid-state lidar architectures utilize dToF sensors, a flash lidar architecture is adopted here for proof of concept. A Bosch GLM250VF rangefinder is used to provide ground truth distance for benchmarking. The implemented guided dToF system is pictured in Figure 9.

## 4. Results

This section evaluates the performance of the guided lidar demonstrator. A visual evaluation assesses the resulting point clouds captured under various challenging dynamic scenarios. This is followed by a quantitative evaluation of the system, including a comparison of the laser power reduction to equivalent partial histogram approaches.

### 4.1. Scenes

#### 4.1.1. Outdoor Clear Conditions

The first scene is conducted under daylight conditions of 15 klux and captures a van driving away from the guided lidar setup. The constituent parts in one frame of guided lidar data from this scene are shown in Figure 10. By configuring the lidar sensor histogram window step size (1.875 m) to be less than the window size (3.12 m), the depth map across the van is continuous even though it spans multiple time windows.

Figure 11 shows the subsequent frames captured from the same scene. The histogram and guided time window of a sample lidar pixel are provided to validate that the pixel is correctly updated as the van drives away. The guided lidar setup continues to track and resolve the distance to the van all the way out to 75 m.

#### 4.1.2. Outdoor Foggy Conditions

Fog presents adverse weather conditions for lidar. Not only does it reduce the intensity of the returning laser, but it also produces early laser returns reflecting from the fog itself [29]. The scene presented in Figure 12 is captured under foggy conditions with both a pedestrian and a car moving separately. The figure shows the time window of a lidar pixel on the pedestrian being correctly updated independently of the pixel on the car, with the car distance resolved as far as 60 m under these challenging conditions.

#### 4.1.3. Transparent Obstacles

Transparent objects such as glass present additional challenges to lidar due to the multipath reflections they introduce [30]. This is particularly problematic for approaches such as partial histogram zooming which favor the first signal peak. The point cloud in Figure 13b shows the result of evaluating only the first peak when presented with a scene through a glass door (Figure 13a). Using a guided dToF approach, each lidar pixel can be correctly guided to the human figure behind the glass door, as shown in Figure 13c.

### 4.2. Performance

#### 4.2.1. Measurement Error

To quantitively evaluate the performance of the guided dToF demonstrator, the measured distance to a human target is compared to the ground truth distance from the rangefinder. A window of 3 × 3 pixels across nine frames is assessed to provide a total of 81 sample points at each distance step. The experiment was conducted outdoors under daylight conditions of 72 klux. As before, the setup is configured to run at 3 fps (no frame averaging). The results are presented in Figure 14, showing the guided lidar maintains a root-mean-squared (RMS) error of less than 20 cm as far 50 m. The error of the guiding stereo depth estimation is also evaluated, showing the squared increase in distance error characteristic of this approach.

#### 4.2.2. Processing Time

The time consumed by each step within a single frame of our guided lidar system is shown in Figure 15. Aside from the lidar acquisition period, the main processes consume a total of 150 ms running on the 1.9 GHz Intel Core i7 processor, limiting the maximum achievable frame rate of this demonstrator to just over 6 fps.

### 4.3. Laser Power Efficiency

To further benchmark the presented guided dToF lidar system, the lidar photon budget (signal and background photon arrival rate) is characterized. This allows established models [21,28] to be applied and the additional laser power consumed by equivalent partial histogram approaches to be determined.

#### 4.3.1. Lidar Characterisation

To characterize the photon budget of our lidar system, a 1 m^2^ Lambertian target calibrated to 10% reflectivity is positioned at various distance intervals and captured. Characterization was performed under ambient daylight conditions of 60 klux. A photograph of the scene during characterization is shown in Figure 16a.

For the signal (laser) photon budget, the lidar exposure time is optimized to ensure a high signal count without clipping. For the background photon budget, the laser is disabled. A total of 100 frames are averaged and a window of 3 × 3 pixels is sampled. The results are shown in Figure 16b. The observed background return rate is independent of distance, in keeping with the literature [31], and is measured to be 4.8 Mcounts/s, equivalent to 8 Mcounts/s at 100 klux. While the observed signal photon return rate varies with distance, it can be considered to follow an inverse square law. Fitting a trendline to this relationship allows the expected signal photons per laser cycle for any target distance (z) for this lidar system to be approximated as 89$z^{-2}$

#### 4.3.2. Laser Power Penalty of Partial Histogram Equivalent

Having characterized the lidar photon budget, the required number of laser cycles as a function of distance for equivalent partial histogram approaches can be quantified. The Thompson model presented in [28] calculates the minimum number of laser cycles for a dToF lidar system to achieve a specified precision. Using the attributes of our lidar sensor (laser pulse and histogram bin width) in Table 2, and the measured photon return rate in Figure 16b, the minimum laser cycles required to achieve 10 cm precision using a full histogram approach as given by the Thompson model is shown in Figure 17a.

An equivalent sliding partial histogram approach (8 × 0.39 m bins sliding in intervals of 3.12 m) would require the same number of laser cycles for distances up to the width of the first slide window. Past this distance, the total laser cycles required to measure any given distance with 10 cm precision is the sum of the full histogram laser cycle value for each additional 3.12 m slide window. The resulting increase in laser cycles for an equivalent sliding partial histogram approach is shown in Figure 17a.

An equivalent zooming partial histogram approach would require an additional zoom step to measure distances greater than 3.12 m by configuring each bin to be 3.12 m wide. Past 25 m, yet another zoom step would be required, configuring each bin to 25 m wide. At each zoom step, the laser must be cycled enough times to detect the peak bin within a specified probability of detection for a given distance. Using the probability of detection model for histogram-based dToF published in [21] and specifying a minimum 99.7% detection rate (3σ rule), the minimum number of laser cycles for each step of an equivalent zooming dToF sensor is given in Figure 17b. The total laser cycles required to zoom to a given distance is therefore the sum of laser cycles required at each zoom step for a given distance, shown in Figure 17a.

The required increase in total laser cycles for each equivalent partial histogram approach compared to a full histogram/guided approach is shown in Figure 17c. At the maximum distance of 75 m achieved by our guided dToF system, a minimum of 6× laser power is saved compared to adopting an equivalent partial histogram approach. It should be noted that the sliding approach modelled here assumes no overlapping of windows between steps which would further increase its laser power penalty.

## 5. Discussion

This section provides a discussion around the evaluated performance of the guided dToF sensor. This is extended to include practical challenges of the presented implementation as well as future work to develop this research further.

### 5.1. Performance Overview

A summary of the presented guided dToF demonstrator performance alongside state-of-the-art dToF lidar sensors is presented in Table 3. The table shows that, while the implemented system uses a single-tier sensor chip with a relatively small number of histogram bins in each macropixel, the combined range and frame rate achieved in bright ambient conditions through a guided dToF approach is amongst the top performing. Using only eight histogram bins per macropixel, the guided dToF demonstrator achieves a maximum distance of 75 m. A conventional full histogram approach with equivalent 0.39 m bin width would require almost 200 histogram bins per macropixel. This is much more than any state-of-the-art sensor is yet to achieve and equivalent to a 24× increase in pixel histogram area for our sensor.

The ability to correctly guide lidar under multipath conditions is also of unique value. In addition to the glass obstruction tested in Figure 15, many other real-world conditions create multiple signal peaks which can be incorrectly interpreted by a standalone lidar. These include obstructions from smoke [32] and retroreflectors such as road signs [33].

**Table 3 sensors-23-08943-t003:** Performance overview of state-of-the-art direct time-of-flight lidar sensors.

Author	Ref	Resolution (ToF pixels)	Max. Range (m)	Ambient Intensity (klux)	Precision/Accuracy (m)	Frame Rate (Hz)	Histogram Bins	Stacked	Partial Histogram
Ximenes	[17]	64 × 128	300 ^1^	-	0.47/0.8	30 ^2^	-	Yes	No
Henderson	[10]	64 × 64	50	-	-/0.17	30	16	Yes	No
Zhang	[9]	144 × 252	50	-	0.0014/0.88	30 ^2^	8	No	Zooming
Okino	[34]	900 × 1200	250	-	1.5/-	-	-	No	No
Kim	[11]	40 × 48	45	-	0.014/0.023	-	2	No	Zooming
Kumagai	[35]	63 × 168	200	117	-/0.3	20	-	Yes	No
Padmanabhan	[18]	128 × 256	100 ^1^	10	-/0.07	-	-	Yes	No
Stoppa	[13]	60 × 80	4.4	50	0.007/0.04	30	32	Yes	Sliding
Zhang	[14]	160 × 240	9.5	10	0.01/0.02	20	32	Yes	Zooming
Park	[15]	60 × 80	45	30	0.015/0.025	1.5 ^2^	4	No	Zooming
Taloud	[16]	32 × 42	8.2	1	0.007/0.03	30	59	Yes	Sliding
This work	-	32 × 64	75	70	0.18	3	8	No	No

^1^ Single point measurement. ^2^ Frame rate not specified at maximum distance.

### 5.2. Design Trade-Offs

By configuring the dToF sensor to use wider depth windows, a greater error can be tolerated from the source of depth estimation. As a result, less accurate but faster stereo algorithms [36] can be used, boosting the system frame rate. This would be particularly significant in this implementation, where stereo depth estimation is the next most time-dominant process after the lidar exposure time (Figure 15). While this could be achieved by increasing the bin width of each dToF pixel histogram, this would be at the cost of increased error in the measured time-of-flight. However, dToF sensors with a greater capacity for histogram storage can take advantage of this by integrating additional bins into each histogram. 

### 5.3. Practical Challenges

For the practical adoption of guided dToF in self-driving vehicles using the specific implementation presented here, the main challenge is in camera misalignment. Any variation in the extrinsic properties of the cameras due to vehicle movement or vibrations not only degrades the stereo depth accuracy but also impacts pixel mapping of depth estimates to the lidar sensor. While continuous camera self-calibration techniques have been developed [37], these need to be explored in the context of a stereo camera-guided dToF lidar system. 

### 5.4. Future Work

Various enhancements are proposed to enable the preliminary guided dToF demonstrator presented here to achieve the performance required for automotive lidar of 200 m range at 25 fps [38]. Firstly, by adopting the increased sensitivity of state-of-the-art SPAD processes [35,39], the maximum range of the system can be extended while reducing the required lidar exposure time. Parallel execution of the various guided processes and adoption of GPU processing will also enable further acceleration to boost frame rate. Finally, the merits of alternative guiding depth sources should be explored, including radar and ultrasound, as well as processing methods such as Kalman filters to extrapolate previous depth frames.

## 6. Conclusions

This paper presents the implementation and evaluation of the first guided dToF system. Our implementation adopts a flash lidar imaging system with stereo depth estimation used to guide the dToF sensor, demonstrating a viable approach to long-range depth sensing through efficient capture of the emitted laser with minimal on-chip data. A guided dToF approach is particularly useful in applications such as self-driving vehicles where a wealth of coarse depth data is available.

## Figures and Tables

**Figure 1 sensors-23-08943-f001:**
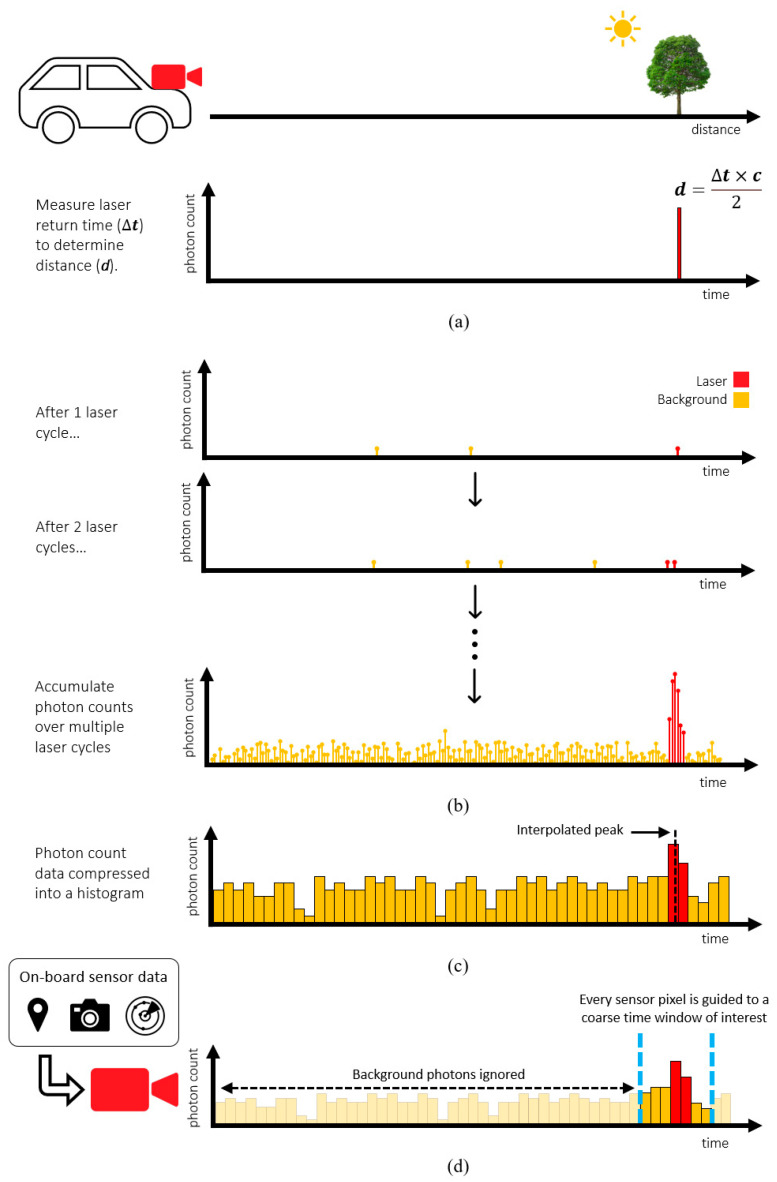
Direct time-of-flight (**a**) ideal case with no ambient background photons (**b**) accumulating photons over multiple laser cycles to average out background photons (**c**) compressing photon data into a histogram (**d**) proposed ‘guided’ approach.

**Figure 2 sensors-23-08943-f002:**
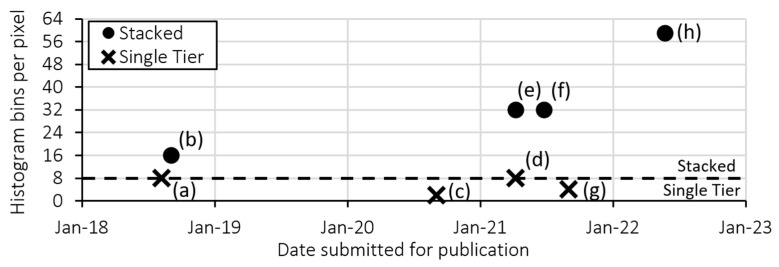
Histogram bin capacity per pixel of published sensors with 2D dToF pixel arrays. (**a**) Zhang 2019 [9], (**b**) Henderson 2019 [10], (**c**) Kim 2021 [11], (**d**) Gyongy 2021 [12], (**e**) Stoppa 2021 [13], (**f**) Zhang 2021 [14], (**g**) Park 2022 [15], (**h**) Taloud (2022) [16].

**Figure 3 sensors-23-08943-f003:**
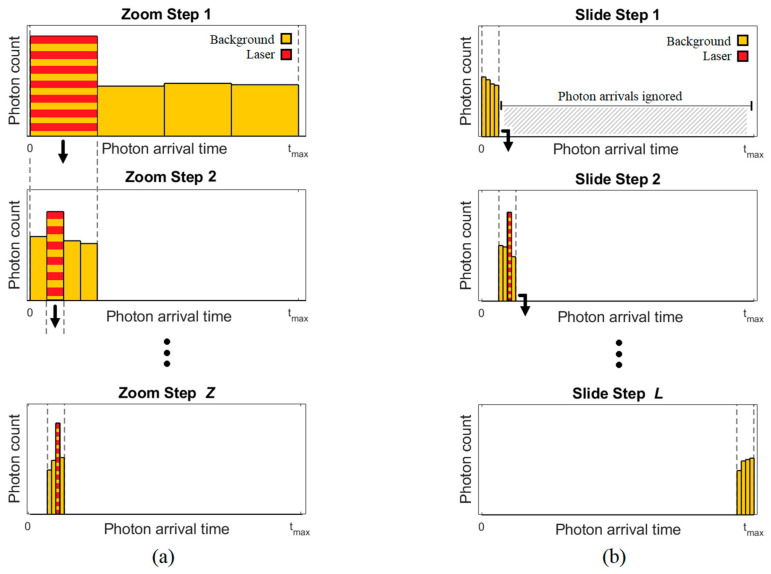
Illustration of partial histogram approaches (**a**) zooming and (**b**) sliding. Reproduced from [19].

**Figure 4 sensors-23-08943-f004:**
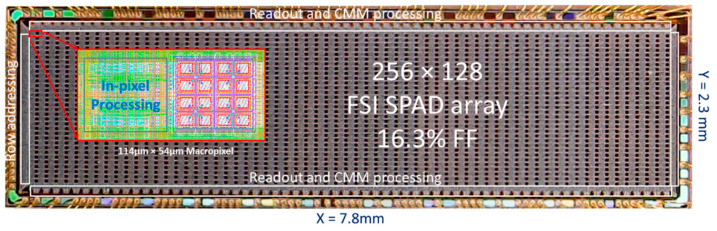
Micrograph of the guided dToF sensor. Reproduced from [12] with author’s permission.

**Figure 5 sensors-23-08943-f005:**
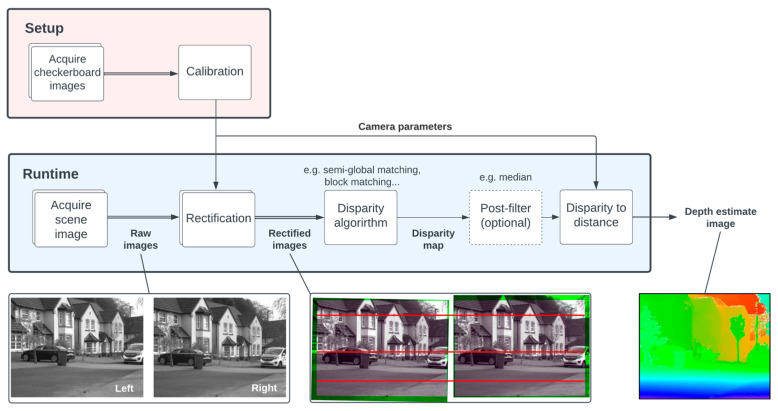
Process flow diagram of acquiring stereo depth estimates for the guided lidar system.

**Figure 6 sensors-23-08943-f006:**
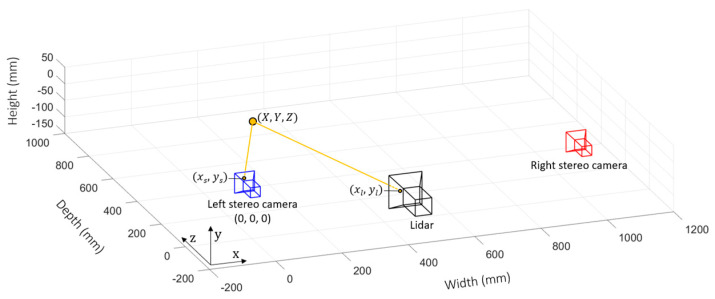
Pixel mapping from the principal stereo camera to the lidar sensor.

**Figure 7 sensors-23-08943-f007:**
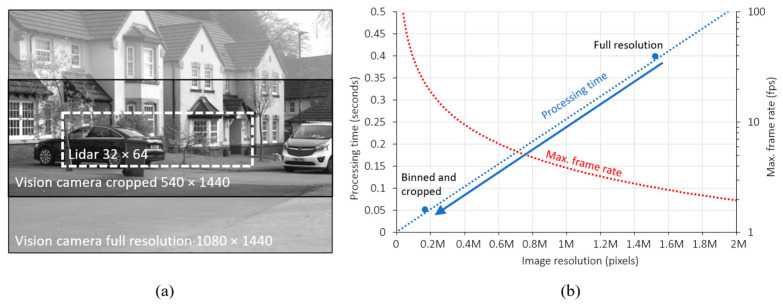
(**a**) Full resolution stereo camera image compared to the projected lidar FOV (**b**) processing time for rectification and SGM stereo depth estimation using the current setup.

**Figure 8 sensors-23-08943-f008:**
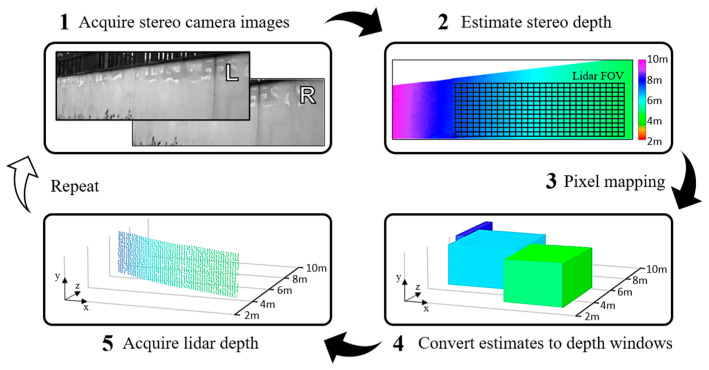
The main steps in the guided dToF lidar process. Adapted from [19].

**Figure 9 sensors-23-08943-f009:**
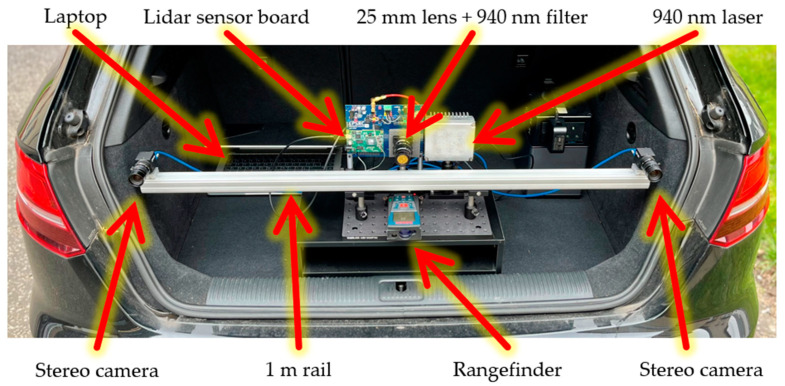
The guided dToF lidar demonstrator described in this publication. Adapted from [19].

**Figure 10 sensors-23-08943-f010:**
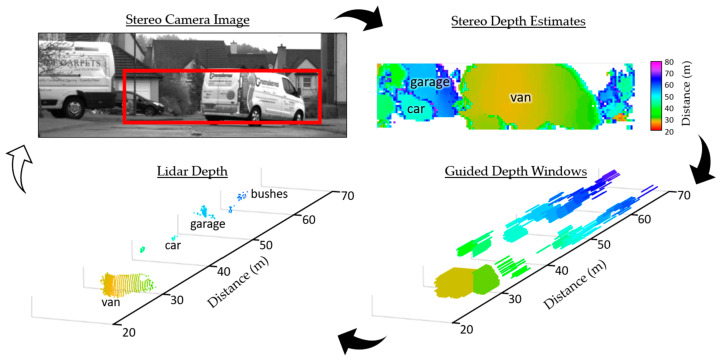
The main process in a single guided dToF frame. Adapted from [19].

**Figure 11 sensors-23-08943-f011:**
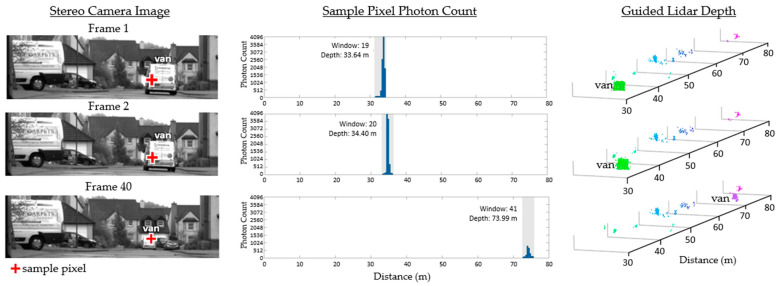
Three further frames of the outdoor scene in Figure 10 on a clear day. A sample dToF sensor pixel shows its configured time window and resulting histogram. Reproduced from [19].

**Figure 12 sensors-23-08943-f012:**
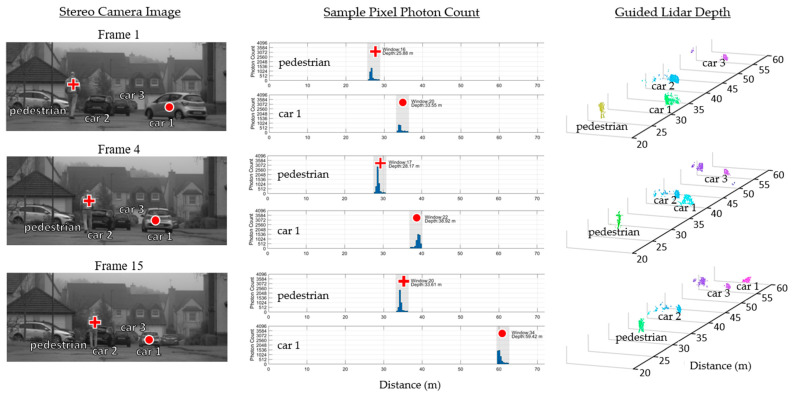
Three frames of an outdoor scene on a foggy day. Sample pixels of the lidar sensor show the time windows they are configured to and the resulting histogram produced.

**Figure 13 sensors-23-08943-f013:**
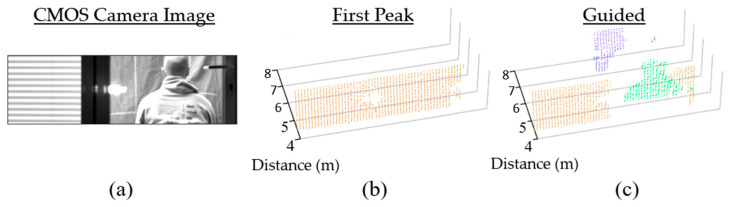
(**a**) a scene through a glass door (**b**) human figure obscured if only the first lidar peak is used (**c**) human figure resolved using guided dToF lidar. Reproduced from [19].

**Figure 14 sensors-23-08943-f014:**
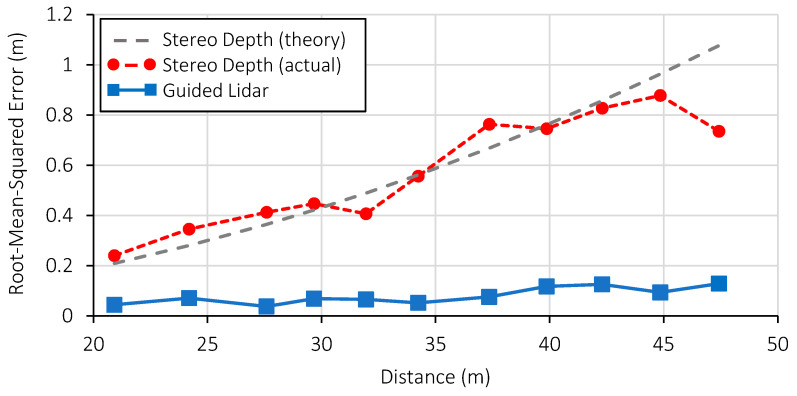
Distance measurement error operating at 3 fps under 72 klux ambient intensity. Theoretical stereo depth accuracy is based on Equation (2), assuming a sub-pixel disparity resolution of 0.25 pixels. Reproduced from [19].

**Figure 15 sensors-23-08943-f015:**
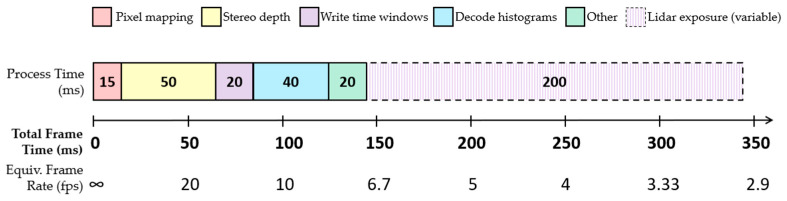
Execution time of each main process in one frame of our guided dToF demonstrator.

**Figure 16 sensors-23-08943-f016:**
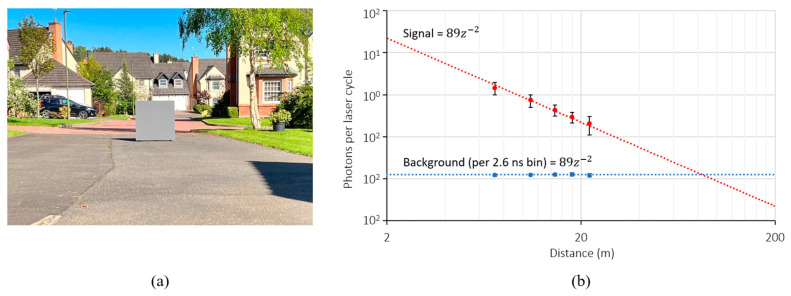
(**a**) Setup during characterization using a 10% reflectivity target under ambient conditions of 60 klux (**b**) measured and extrapolated photon return rate during characterization.

**Figure 17 sensors-23-08943-f017:**
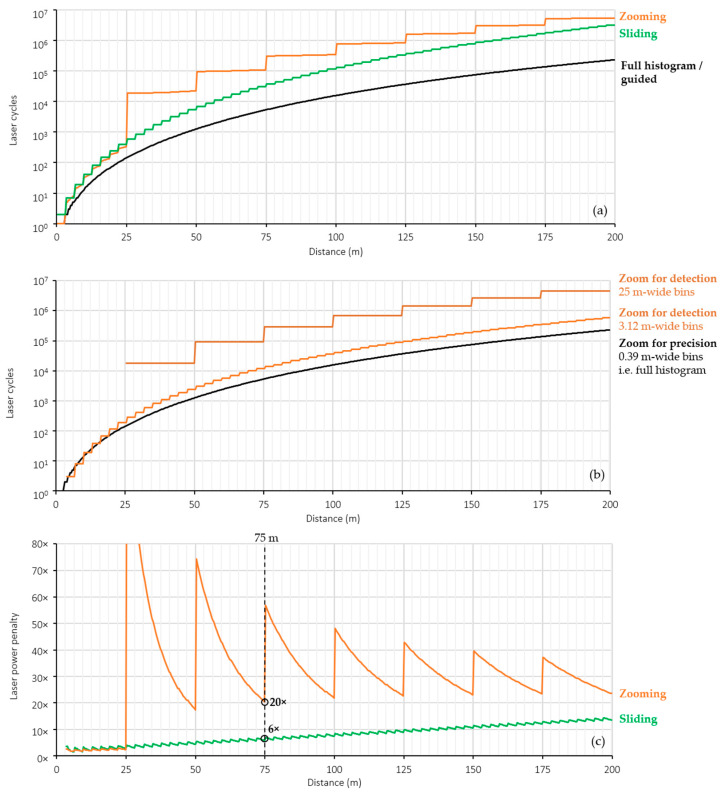
(**a**) Minimum laser cycles required to achieve 10 cm depth precision using the presented guided dToF configuration and equivalent sliding and zooming partial histogram approaches (**b**) minimum laser cycles required for each step of an equivalent zooming partial histogram approach (**c**) laser power penalty of equivalent sliding and zooming partial histogram approaches.

**Table 1 sensors-23-08943-t001:** Merits of different dToF histogram approaches alongside the proposed guided approach.

Parameter	Full Histogram	Partial Histogram		Guided
		Zooming	Sliding	
Laser power penalty	Low	High	High	Low
Area requirement	High	Low	Low	Low
Data volume	High	Low	High	Low
Multipath reflection artefacts	Low	Medium	Low	Low
Motion artefacts	Low	Medium	Medium	Medium
System complexity	Low	Low	Low	High

**Table 2 sensors-23-08943-t002:** Component and attributes for the presented guided dToF lidar system.

Component	Parameter	Value
Stereo Rig	Baseline	1 m
	Camera model	FLIR BFS-U3-16S2M-CS
	Maximum resolution	1080 × 1440
	Focal length	12 mm
Lidar	Laser pulse width	4.5 ns FWHM
	Laser repetition rate	80 kHz
	Wavelength	940 nm
	Filter bandwidth	10 nm FWHM
	Focal length	25 mm
	Field of view (H × V)	16° × 4°
	Histogram bins	8 × 12-bit
	Histogram bin width	0.39 m (2.6 ns)
	Histogram window step	1.875 m (1.25 ns)

## Data Availability

Not applicable.

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
