# Peer review of "Guided Direct Time-of-Flight Lidar Using Stereo Cameras for Enhanced Laser Power Efficiency"

_sensors, 2023, doi:10.3390/s23218943_

Round 1
Reviewer 1 Report
Comments and Suggestions for Authors
The research work is interesting. Congratulations. However, some comments to improve the work are provided below.
General Comments.
- Mathematical equations should be part of the text. Therefore, it should end with a comma or period.
- Add an introduction to each section. Explain shortly what each section is about.
- Include a section on conclusions and future work.
Author Response
Q1: Mathematical equations should be part of the text. Therefore, it should end with a comma or period.
A1: Equation (3), (4) and (5) follow the end of a sentence and have been punctuated as per reviewer’s recommendation.
Q2: Add an introduction to each section. Explain shortly what each section is about.
A2: An introductory paragraph (highlighted in revised manuscript) has been added to each section (apart from the first section “1. Introduction”).
Q3: Include a section on conclusions and future work.
A3: The discussion now has a dedicated subsection “5.4 Future Work” and a new “6. Conclusion” section has been added.
Reviewer 2 Report
Comments and Suggestions for Authors
Software tools and specific scripts used for tests should be described.
Author Response
Q1: Software tools and specific scripts used for tests should be described
A1: A new section “3.6 Software” has been added to address this.
Reviewer 3 Report
Comments and Suggestions for Authors
The paper is well written. However, the authors should strengthen more the originality of the paper. Some comments
1) which is the response time?
2) how different tuning of the parameters affect the whole process?
3) which is the main drawbacks of your method
Author Response
Q1: which is the response time?
A1: Section 4.2.2 documents the processing time of the system. 150 ms pass at the start of each frame before the lidar acquisition begins.
Q2: how different tuning of the parameters affect the whole process?
A2: A new section “5.2 Design Trade-Offs” has been added to address this.